# Early Developmental Signs in Children with Autism Spectrum Disorder: Results from the Japan Environment and Children’s Study

**DOI:** 10.3390/children9010090

**Published:** 2022-01-10

**Authors:** Hideki Shimomura, Hideki Hasunuma, Sachi Tokunaga, Yohei Taniguchi, Naoko Taniguchi, Tetsuro Fujino, Takeshi Utsunomiya, Yasuhiko Tanaka, Narumi Tokuda, Masumi Okuda, Masayuki Shima, Yasuhiro Takeshima

**Affiliations:** 1Department of Pediatrics, Hyogo College of Medicine, Nishinomiya 663-8501, Japan; sa-tokunaga@hyo-med.ac.jp (S.T.); yo-taniguchi@hyo-med.ac.jp (Y.T.); nao-taniguchi@hyo-med.ac.jp (N.T.); tet-ped@hyo-med.ac.jp (T.F.); ta-utsunomiya@hyo-med.ac.jp (T.U.); tana-ped@hyo-med.ac.jp (Y.T.); mash0807@hyo-med.ac.jp (M.O.); ytake@hyo-med.ac.jp (Y.T.); 2Hyogo Regional Center for the Japan Environment and Children’s Study, Hyogo College of Medicine, Nishinomiya 663-8501, Japan; hi-hasunuma@hyo-med.ac.jp (H.H.); na-tokuda@hyo-med.ac.jp (N.T.); shima-m@hyo-med.ac.jp (M.S.); 3Department of Public Health, Hyogo College of Medicine, Nishinomiya 663-8501, Japan

**Keywords:** autism spectrum disorder, early signs, ASQ-3

## Abstract

Autism spectrum disorder (ASD) is a developmental disability in early childhood. Early identification and intervention in children with ASD are essential for children and their families. This study aimed to identify the earliest signs of ASD. Using a large cohort including data from 104,062 fetal records in the Japan Environment and Children’s Study, we examined the Ages and Stages Questionnaires^®^ (ASQ-3^TM^) scores of children with and without ASD. The ASQ-3 comprises five domains: communication, gross motor, fine motor, problem solving, and personal-social. The ASQ-3 scores were obtained at ages 6 months, 1 year, and 3 years. There were 64,501 children with available ASQ-3 data. The number of children diagnosed with ASD was 188 (0.29%) at 3 years of age. The highest relative risk (RR) for any domain below the monitoring score at 6 months was in the communication (RR 1.90, 95% CI 1.29–2.78, *p* = 0.0041), followed by fine motor (RR 1.50, 95% CI 1.28–1.76, *p* < 0.0001) domain. A low ASQ-3 score in the communication domain at 6 months was related to an ASD diagnosis at 3 years of age. The ASQ-3 score at 6 months can contribute to the early identification of and intervention for ASD.

## 1. Introduction

Autism spectrum disorder (ASD) is a developmental disability characterized by deficits in social interaction, restricted interests, and repetitive behaviors, which manifest early in life [1]. A recent study reported that the prevalence of ASD has increased, and it is currently estimated to range from 13.1 to 29.3 per 1000 children [2]. This prevalence varies according to sex and race/ethnicity [3]. The prevalence among Japanese children was 2.1 per 1000 in 1996 [4] and 19 per 1000 children in 2015 [5]. Although these reports cannot be directly compared because different participants and survey methods were involved, this increasing prevalence is consistent with previous reports in other countries [2].

Early identification and intervention in children with ASD are essential to improve the abilities, health outcomes, and quality of life of the children and their families [6]. It is not feasible to perform a detailed examination of all children who may be considered for ASD diagnosis since the number of cases is too large. In Japan, public health checkups are conducted at ages 4, 10, and 18 months, and we believe that these are crucial periods for ASD screening. We must know what signs we should focus on to identify ASD efficiently in the health checkups. Early cognitive and motor development signs of ASD have been reported in previous studies [7,8,9]. However, there is still a debate on which sign appears earlier.

Several studies have investigated the relationship of genetic [10] and environmental factors, such as parental age, underlying illness, and drugs, with the development of ASD [11]. Advanced parental age, especially paternal age, has been reported as one of the most significant risk factors for ASD [12]. In the case of advanced paternal age, one mediating mechanism could be the impairment of neuronal development due to the formation of de novo mutations in germline cells and modifications in DNA methylation [13]. In contrast, advanced maternal age affects chromosomal abnormalities [14], which might result in the development of ASD. Regarding physical disorders, those such as tic [15], neurofibromatosis type 1 [16], and Duchenne muscular dystrophy [17] have been suggested to be associated with ASD. In addition, research for medical conditions in infancy as predictive markers for later ASD diagnosis revealed that generalized convulsive epilepsy, nystagmus, and strabismus were likely to be associated with a later ASD diagnosis [18].

The Ages and Stages Questionnaires Third Edition (ASQ-3) is a parent-completed developmental screening tool used to identify potential developmental delays in children [19]. The ASQ-3 is a low-cost and simple tool with high reliability and validity, which has been implemented in several countries, including Japan, and is often used in research and clinical practice [19]. It is considered a suitable tool for assessing the development of primary disease. This study aimed to identify the earliest signs of ASD in children without intellectual disability using the ASQ-3.

## 2. Materials and Methods

In this study, we used data from a large-scale cohort study, the Japan Environment and Children’s Study (JECS), which is a government-funded birth cohort study that started in January 2011 to elucidate the effects of environmental factors on maternal and infant health [20,21]. In the JECS, 104,062 fetal records were extracted across the 15 regional centers, which are representative of the diverse Japanese social, economic, and urban realities during a 3-year period that ended in March 2014. The JECS protocol was reviewed and approved by the Ministry of the Environment’s Institutional Review Board (approved 10 September 2010) on Epidemiological Studies and the Ethics Committees of all participating institutions. Written informed consent was obtained from all participants. This study was funded by the Ministry of the Environment, Japan.

### 2.1. Participants

The study cohort was extracted from dataset “jecs-ta-20190930-add001”, which was released in October 2019 and has been described in detail elsewhere [21]. This dataset included 104,062 fetal records and 100,304 live birth records. From these 100,304 live birth records, we excluded participants whose answers in the ASQ-3 (at ages 6 months, 1 year, and 3 years) were not completed (*n* = 35,336). To identify the early signs of ASD, children diagnosed with motor delay and/or intellectual disability (motor delay, 155; intellectual disability, 440; overlapping, 126) were excluded. This is because developmental delay is recognized even if the child has intellectual disability and motor delay. The diagnosis of ASD, motor delay, and intellectual disability was based on questionnaires completed by the caregiver. The question includes whether your child had received the above-mentioned diagnosis from a doctor in the past. In total, 64,501 children were included in the analysis. To assess the effect of paternal age, we exclude participants whose father’s age is missing (*n* = 29,605), the corresponding data were extracted (*n* = 34,896), and a stratified analysis was performed (Figure 1).

### 2.2. Data Selection of the ASQ-3

The ASQ-3 scores were obtained from the caregiver-completed questionnaires for all children (*n* = 64,499). Children were then categorized into those who were diagnosed with ASD at 3 years of age and those who were not.

The ASQ-3 comprised 21 age-specific questionnaires for children aged 1–66 months to assess their development in five domains. The domains included communication, gross motor, fine motor, problem solving, and personal-social [22]. Each of the five domains contained six questions; thus, 30 items were included in each questionnaire. The caregivers completed the questionnaires. There were three possible answers for each question: “yes” when the behavior was present (10 points), “sometimes” when the behavior was emerging (5 points), or “not yet” when the behavior was absent (0 points). The caregiver could omit items when they were unsure of how to respond or due to concerns about the child’s performance. The ASQ-3 scores were not calculated if there were three or more omitted items in a given domain. In the case of one or two omitted items, an adjusted total domain score was calculated by adding the average item score either once for one omission or twice for two omissions. To calculate the ASQ-3 scores for each domain, the score of all items in the corresponding domain was added.

The domain-specific ASQ-3 scores were compared with the Japanese validated score [23]. In the Japanese validated score, the monitoring and cut-off scores were defined as 1 and 2 standard deviations below the mean score, respectively. Based on this, the participants were categorized into groups below the monitoring and below the cut-off scores. The relative risk (RR) was calculated in each domain in children with ASD compared to children without ASD. Considering that paternal age can influence the development of ASD [10,11,12], we also stratified the children according to a paternal age cut-off value of 35 years.

### 2.3. Statistical Analysis

Statistical analysis was performed using JMP Pro 14 software for Windows (SAS Institute Inc., Cary, NC, USA). Comparisons of the number of children below the monitoring score and cut-off score were performed using Fisher’s exact test. The level of significance was set at *p* < 0.05.

## 3. Results

The number of children diagnosed as having ASD without an intellectual disability was 188 (0.29%), and there was a male predominance (143 boys and 45 girls).

Table 1 displays the distribution of children whose results fall below the monitoring and cut-off scores. In the whole-sample analysis, the gross motor domain was most frequently observed to be below the monitoring score at the 6-month and 1-year evaluations, and personal-social was most frequently observed to be below the monitoring score at the 3-year evaluation. Problem solving was most frequently observed to be below the cut-off score at the 6-month evaluation, while fine motor was the most affected at the 1-year and 3-year evaluations. In the subset of children with ASD, fine motor was most frequently observed to be below the monitoring score at the 6-month evaluation, and personal-social was the most frequently observed domain to be below the monitoring score at the 1-year and 3-year evaluations.

The RR at 6 months was the highest in children in whom the communication domain was below the monitoring score (RR 1.90, 95% confidence interval (CI) 1.29–2.78, *p* = 0.0041) and below the cut-off score (RR 2.49, 95% CI 0.81–7.69, *p* = 0.1218), and the second-highest RR was observed in the fine motor domain (below the monitoring score: RR 1.50, 95% CI 1.28–1.76, *p* < 0.0001; below the cut-off score: RR 2.20, 95% CI 1.48–3.26, *p* = 0.005). At the 1-year evaluation, the highest RR was seen when the communication domain was below the monitoring score (RR 3.93, 95% CI 3.11–4.96, *p* < 0.0001), and the second-highest RR was seen in the personal-social domain (RR 2.57, 95% CI 2.18–3.02, *p* < 0.0001). At the 3-year evaluation, the domain that showed the highest RR when below the monitoring score was communication (RR 6.54, 95% CI 5.97–7.16, *p* < 0.0001), followed by problem solving (RR 4.50, 95% CI 4.11–4.94, *p* < 0.0001).

The results of stratification based on paternal age are shown in Table 2. The number of children whose fathers were under 35 years of age was 20,830, while 14,065 children had fathers aged 35 years and above. At the age of 6 months, communication was the domain with the highest RR (RR 2.61, 95% CI 1.24–5.50, *p* = 0.026) when below the monitoring score in the group with fathers under 35 years of age, and the second-highest RR was observed in the personal-social domain (RR 1.52, 95% CI 1.01–2.28, *p* = 0.071). In children with fathers aged 35 years and above, communication also had the highest RR (RR 1.90, 95% CI 1.00–3.60, *p* = 0.070) at the same period when below the monitoring score, followed by the fine motor domain (RR 1.58, 95% CI 1.22–2.04, *p* = 0.005). At the 1-year evaluation, the highest RR was observed when the communication domain was below the monitoring score (under 35 years old: RR 4.87, 95% CI 3.08–7.69, *p* < 0.0001; 35 years old and above: RR 3.61, 95% CI 2.39–5.47, *p* < 0.0001), and the second-highest RR was observed in the personal-social domain (under 35 years old: RR 2.45, 95% CI 1.66–3.61, *p* = 0.0004; 35 years old and above: RR 2.27, 95% CI 1.68–3.06, *p* < 0.0001) in both paternal age groups. Finally, at the 3-year evaluation, the highest RR for a domain below the monitoring score was for communication (RR 8.34, 95% CI 7.19–9.68, *p* < 0.0001), and the second-highest RR was for problem solving (RR 4.97, 95% CI 4.11–6.00, *p* < 0.0001). In children with fathers aged 35 years and above, the highest RR was also seen in the communication domain (RR 4.77, 95% CI 3.81–5.98, *p* < 0.0001), and the second-highest RR was in the gross motor domain (RR 4.03, 95% CI 3.07–5.28, *p* < 0.0001).

## 4. Discussion

This study aimed to identify the earliest signs of ASD. Using data from a large cohort study, we revealed that children at 6 months of age who scored below the monitoring range for the domain of communication or fine motor had a greater relative risk of developing ASD by 3 years of age. Thus, the earliest features of ASD at 6 months were impaired communication and fine motor skills. The results of stratification based on paternal age demonstrated that children at 6 months of age who scored below the monitoring range for the domain of communication had a greater relative risk of developing ASD by 3 years of age. This result is similar to unstratified participants seen at 6 months of age.

Early signs of ASD have been reported in a considerable number of studies, including language delays [7], impaired visual attention [8], stereotyped behaviors [24], and motor difficulties, both in fine and gross motor skills [9]. Although most studies examining early signs of ASD have focused on language, social behavior, and cognitive development, other studies have also focused on motor function. Recently, the number of video-based studies analyzing spontaneous movement has increased [25]. However, determining the earliest signs of ASD remains controversial [26]. In this study, we found that delays in communication were more noticeable than motor impairments at 6 months of age. This suggests that infants with ASD developed delays in communication at a fairly early age; however, we could not determine which features appeared earlier since the delay in fine and gross motor functions was already present. In order to find high-risk children, it was necessary to use a cut-off value; however, in this study, the monitoring value was used. This was due to the low number of children whose scores fall below the cut-off value despite the inclusion of a large number of participants, which prevented the corresponding statistical testing.

According to the results of the stratified analysis, although the father’s age is known to affect the incidence of ASD, similar to the unstratified participants, the earliest sign was found to be a delay in communication. However, especially in the stratified analysis, since the number of children diagnosed with ASD in each group was small, this finding should be interpreted carefully. Compared to the overall sample, the second earliest signs were different at the 6-month evaluation in the group of children with fathers under 35 years old; personal-social skills showed the second greatest impairment. Further, at the 3-years evaluation in the group of children with fathers aged 35 years and above, gross motor skills showed the second greatest impairment. Thus, in children with older fathers, motor signs may be more evident in children with ASD; however, excluding children whose father’s age was unknown may have skewed the results.

In order to promote the early identification of early signs of ASD, recommendations have been published in several countries [26,27]. Although several tests other than the Ages and Stages Questionnaires Third Edition (ASQ-3) have been used to identify ASD, the modified checklist for autism in toddlers (M-CHAT) was the most commonly used [28,29].

Several studies have shown the usefulness of ASQ-3 as a screening tool for ASD [19,30,31]. A cross-sectional study [28] of children with ages ranging from 16 to 30 months revealed that the communication domain of the ASQ-3 identified most children being diagnosed with ASD based on the M-CHAT form. However, the ASQ-3 sensitivity for the detection of children screened using M-CHAT was relatively low (70%). Thus, the authors proposed that a two-stage screening process may be effective for identifying children with ASD. Regarding motor function, Vanvuchelen et al. [30] reported that the ASQ-3 gross and fine motor domains were useful in identifying children with ASD who had unnoticed motor problems at 22 to 54 months. Together, these reports suggest that impairments in communication and motor function were conspicuous signs after 16 months. In our study, impairments in communication were most evident at this time, as in previous reports [7,31]. However, the second and third most conspicuous signs are not impairments in motor function but personal-social skills and problem solving. This difference might be due to cultural differences in development. The effect of cultural differences on the ASQ score was also recognized when the validated scores were analyzed [23], and it is thought that the same was recognized in our study. Regarding the ASD screening, previous reports [30,31] recommended the addition of a formal standardized motor test in the diagnostic procedure. In our study, it is suggested that children at 6 months of age who scored below the monitoring range for the domain of communication in ASQ–3 had a greater relative risk of developing ASD by 3 years of age. Thus, we suggest that it is reasonable to screen children under 18 months of age to identify those at a higher risk, and various tests, including M-CHAT, should be performed at 18 months of age. However, further examination is needed to consider whether ASQ-3 can be used as a screening tool.

Identifying early signs of ASD is considered as essential for examination of traumatized children. This is because autistic traits were often confused with symptoms of post-traumatic stress disorder (PTSD) in later life; therefore, a considerable number of adolescences with PTSD were thought to be diagnosed with ASD [32,33]. On the other hand, it should be also noted that children with ASD have increased risk for encountering traumatic events [34].

This study has some limitations. The major limitation was that the diagnosis of ASD was not confirmed using the same diagnostic criteria. Since the information was obtained using a questionnaire completed by the caregiver, it was difficult to ensure diagnostic consistency. Second, data on the diagnosis of ASD used in this study were collected from questionnaires completed at 3 years of age. It is important to note that a considerable number of children with ASD will be diagnosed after 3 years of age; thus, it is likely that the children without ASD at 3 years of age may have included those who have not yet been diagnosed with ASD. Third, the participants with intellectual disability and motor delay were excluded because this study was focused on the development of pure ASD. Therefore, a certain number of children with ASD with an intellectual disability or motor delay were excluded, and the number of children with ASD who were analyzed in this study was lower. Finally, although various potential risk factors for ASD have been reported, in this study, since the number of children with ASD was limited, a stratified analysis was performed only based on paternal age. Thus, it is necessary to consider the relationship with other factors by analyzing more children. Further studies should be conducted to continue assessing the children in this study and improve the diagnostic system by collecting additional medical information.

## Figures and Tables

**Figure 1 children-09-00090-f001:**
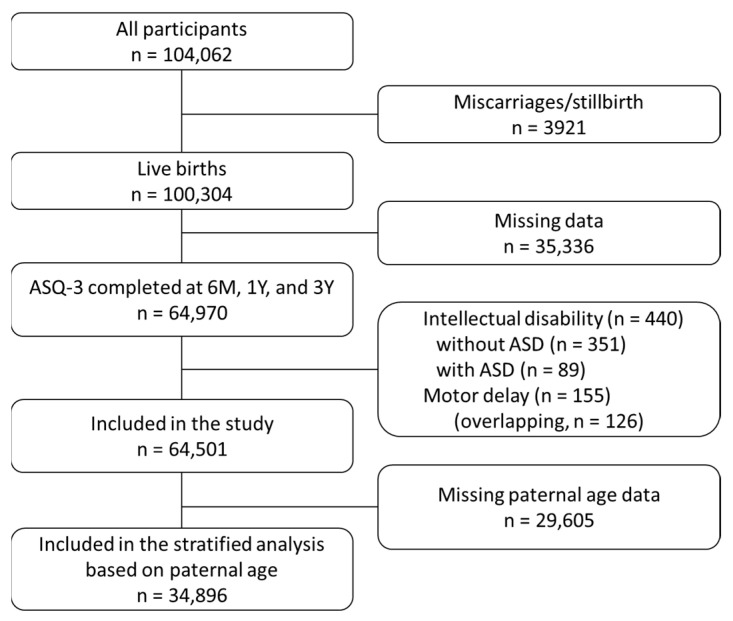
Flowchart of the study participants. ASD, autism spectrum disorder; ASQ-3, Ages and Stages Questionnaires Third Edition; M, months; Y, years.

**Table 1 children-09-00090-t001:** Distribution of children whose results fall below the ASQ-3 monitoring and cut-off scores in each domain.

	Domains		Total	Non-ASD	ASD	RR	95% CI		*p*-Value
6 months old									
	Communication	Monitoring	4171	4148	23	1.90	1.29	2.78	0.0041
		Cut-off	415	412	3	2.49	0.81	7.69	0.1218
	Gross motor	Monitoring	19,813	19,735	78	1.35	1.14	1.60	0.0019
		Cut-off	6914	6878	36	1.79	1.33	2.40	0.0005
	Fine motor	Monitoring	19,228	19,144	84	1.50	1.28	1.76	<0.0001
		Cut-off	3444	3422	22	2.20	1.48	3.26	0.0005
	Problem solving	Monitoring	16,935	16,862	73	1.48	1.24	1.77	0.0002
		Cut-off	7259	7224	35	1.66	1.23	2.24	0.0025
	Personal−social	Monitoring	16,347	16,277	70	1.47	1.22	1.77	0.0003
		Cut-off	2470	2456	14	1.95	1.18	3.23	0.0194
1 year old									
	Communication	Monitoring	4582	4530	52	3.93	3.11	4.96	<0.0001
		Cut-off	77	75	2	9.12	2.26	36.88	0.0214
	Gross motor	Monitoring	12,250	12,187	63	1.77	1.44	2.16	<0.0001
		Cut-off	3660	3634	26	2.45	1.71	3.50	<0.0001
	Fine motor	Monitoring	10,130	10,076	54	1.83	1.46	2.30	<0.0001
		Cut-off	3732	3707	25	2.31	1.60	3.33	0.0001
	Problem solving	Monitoring	10,134	10,065	69	2.35	1.94	2.83	<0.0001
		Cut-off	3340	3308	32	3.31	2.41	4.55	<0.0001
	Personal−social	Monitoring	11,139	11,056	83	2.57	2.18	3.02	<0.0001
		Cut-off	783	770	13	5.78	3.40	9.80	<0.0001
3 years old									
	Communication	Monitoring	7254	7118	136	6.54	5.97	7.16	<0.0001
		Cut-off	2279	2180	99	15.54	13.48	17.90	<0.0001
	Gross motor	Monitoring	7486	7391	95	4.40	3.81	5.07	<0.0001
		Cut-off	2671	2614	57	7.46	5.99	9.29	<0.0001
	Fine motor	Monitoring	10,102	9993	109	3.73	3.30	4.22	<0.0001
		Cut-off	4634	4556	78	5.86	4.93	6.96	<0.0001
	Problem solving	Monitoring	10,311	10,177	134	4.50	4.11	4.94	<0.0001
		Cut-off	4496	4391	105	8.18	7.18	9.32	<0.0001
	Personal−social	Monitoring	11,523	11,374	149	4.48	4.16	4.83	<0.0001
		Cut-off	1849	1760	89	17.30	14.78	20.25	<0.0001

Abbreviations: ASQ-3, Ages and Stages Questionnaires Third Edition; ASD, autism spectrum disorder; RR, risk ratio; CI, confidence interval. Monitoring and cut-off scores are relative to the Japanese validated ASQ-3 score (23) for each domain. The relative risk (RR) was calculated in each domain between children with and without ASD. In the Japanese validated score, the monitoring and cut-off scores were defined as 1 and 2 standard deviations below the mean score, respectively.

**Table 2 children-09-00090-t002:** Stratified analysis based on paternal age.

	Domains		Non-ASD	ASD	RR	95% CI		*p*-Value
Under 35 years old, *n* = 20,830								
6 months old								
	Communication	Monitoring	1111	6	2.61	1.24	5.50	0.026
		Cut-off	99	1	4.88	0.70	34.22	0.187
	Gross motor	Monitoring	6173	16	1.25	0.85	1.85	0.316
		Cut-off	2072	7	1.63	0.83	3.22	0.195
	Fine motor	Monitoring	5736	17	1.43	0.99	2.07	0.088
		Cut-off	915	6	3.17	1.50	6.68	0.011
	Problem solving	Monitoring	4876	10	0.99	0.58	1.71	1.000
		Cut-off	1979	4	0.98	0.38	2.49	1.000
	Personal−social	Monitoring	4780	15	1.52	1.01	2.28	0.071
		Cut-off	664	3	2.18	0.73	6.52	0.158
1 year old								
	Communication	Monitoring	1291	13	4.87	3.08	7.69	<0.0001
		Cut-off	18	1	26.86	3.67	196.73	0.039
	Gross motor	Monitoring	3593	15	2.02	1.34	3.04	0.007
		Cut-off	1010	3	1.44	0.48	4.28	0.466
	Fine motor	Monitoring	3038	15	2.39	1.58	3.60	0.001
		Cut-off	1035	7	3.27	1.66	6.46	0.005
	Problem solving	Monitoring	2971	14	2.28	1.48	3.51	0.003
		Cut-off	870	8	2.28	1.48	3.50	0.003
	Personal−social	Monitoring	3160	16	2.45	1.66	3.61	0.0004
		Cut-off	195	4	9.92	3.86	25.48	0.001
3 years old								
	Communication	Monitoring	2029	35	8.34	7.2	9.7	<0.0001
		Cut-off	592	22	17.96	13.3	24.3	<0.0001
	Gross motor	Monitoring	2208	20	4.38	3.2	6.0	<0.0001
		Cut-off	750	12	7.73	4.8	12.6	<0.0001
	Fine motor	Monitoring	3065	27	4.26	3.4	5.4	<0.0001
		Cut-off	1361	19	6.75	4.8	9.5	<0.0001
	Problem solving	Monitoring	3017	31	4.97	4.1	6.0	<0.0001
		Cut-off	1241	21	8.18	6.0	11.2	<0.0001
	Personal−social	Monitoring	3464	35	4.88	4.2	5.7	<0.0001
		Cut-off	498	19	18.44	13.0	26.1	<0.0001
35 years old and over, *n* = 14,065								
6 months old								
	Communication	Monitoring	1095	8	1.90	1.00	3.60	0.070
		Cut-off	115	1	2.26	0.32	15.86	0.361
	Gross motor	Monitoring	4625	26	1.46	1.10	1.93	0.021
		Cut-off	1711	13	1.97	1.23	3.17	0.019
	Fine motor	Monitoring	4612	28	1.58	1.22	2.04	0.005
		Cut-off	902	7	2.01	1.01	4.03	0.085
	Problem solving	Monitoring	4177	25	1.55	1.16	2.07	0.011
		Cut-off	1831	16	2.27	1.50	3.43	0.002
	Personal−social	Monitoring	3902	22	1.46	1.06	2.02	0.047
		Cut-off	659	3	1.18	0.39	3.56	0.741
1 year old								
	Communication	Monitoring	1149	16	3.61	2.39	5.47	<0.0001
		Cut-off	23	1	11.28	1.55	82.05	0.088
	Gross motor	Monitoring	3003	18	1.56	1.07	2.27	0.045
		Cut-off	923	10	2.81	1.60	4.94	0.003
	Fine motor	Monitoring	2436	13	1.38	0.86	2.23	0.207
		Cut-off	970	7	1.87	0.94	3.75	0.099
	Problem solving	Monitoring	2498	16	1.66	1.10	2.51	0.032
		Cut-off	849	5	1.53	0.66	3.53	0.381
	Personal−social	Monitoring	2747	24	2.27	1.68	3.06	<0.0001
		Cut-off	209	1	1.24	0.18	8.69	0.557
3 years old								
	Communication	Monitoring	1739	32	4.77	3.8	6.0	<0.0001
		Cut-off	566	22	10.09	7.2	14.1	<0.0001
	Gross motor	Monitoring	1740	27	4.03	3.1	5.3	<0.0001
		Cut-off	656	16	6.33	4.2	9.6	<0.0001
	Fine motor	Monitoring	2252	26	3.00	2.3	4.0	<0.0001
		Cut-off	1089	19	4.53	3.1	6.5	<0.0001
	Problem solving	Monitoring	2040	35	3.75	3.1	4.6	<0.0001
		Cut-off	1103	24	5.65	4.2	7.6	<0.0001
	Personal−social	Monitoring	2777	37	3.46	2.9	4.2	<0.0001
		Cut-off	468	22	12.20	8.7	17.0	<0.0001

Abbreviations: ASD, autism spectrum disorder; RR, risk ratio; CI, confidence interval. Monitoring and cut-off scores are relative to the Japanese validated ASQ-3 score (23) for each domain. The relative risk (RR) was calculated in each domain between children with and without ASD. In the Japanese validated score, the monitoring and cut-off scores were defined as 1 and 2 standard deviations below the mean score, respectively.

## Data Availability

Data are unsuitable for public deposition due to ethical restrictions and the legal framework of Japan. It is prohibited by the Act on the Protection of Personal Information (Act No. 57 of 30 May 2003, amendment on 9 September 2015) to publicly deposit the data containing personal information. Ethical Guidelines for Medical and Health Research Involving Human Subjects enforced by the Japan Ministry of Education, Culture, Sports, Science and Technology and the Ministry of Health, Labour and Welfare also restrict the open sharing of the epidemiologic data. All inquiries about access to data should be sent to: jecs-en@nies.go.jp. The person responsible for handling inquiries sent to this e-mail address is Shoji F. Nakayama, JECS Programme Office, National Institute for Environmental Studies.

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
