# Peer review of "Early Developmental Signs in Children with Autism Spectrum Disorder: Results from the Japan Environment and Children’s Study"

_children, 2022, doi:10.3390/children9010090_

Round 1

Reviewer 1 Report

Shimomura et al. investigated the early developmental signs in children with autistic spectrum 2 disorder. There were not so many similar studies in current literature, especially in so large population. The study is well designed. Authors revealed,  that low ASQ-3 score in the communication domain at 6 months 23 was related to an ASD diagnosis at 3 years of age. The paper is well-written. However I have one minor comment for author-I think it would be worth to add short paragraph in introduction section about some disorders (like for example tics- Neurol Neurochir Pol 2019;53(5):315-316.) in which the ASD are frequently observed.

Reviewer 2 Report

The authors have performed a valuable study. There are some comments that can be made:

  1. The discussion should be aligned. Now there is much content in the discussion that belongs to the introduction. I would suggest the discussion to be focused on: what was found (short) and what was not found and how do you explain this/limitations of the study/conclusion and implications for practice/research.
  2. The importance of the study can be higlighted by the observations that signs of autism are often confused with trauma in later life and many traumatised children with PTSD are probably wrongly diagnosed with autism in adolescence. But also autism could lead to trauma in society. ref: Warrier, V., Baron-Cohen, S. Childhood trauma, life-time self-harm, and suicidal behaviour and ideation are associated with polygenic scores for autism. Mol Psychiatry26, 1670–1684 (2021). https://doi.org/10.1038/s41380-019-0550-x
  3. This  study highlights the importance of early screening as recent programs have shown to alleviate signs of autism when applied early.
